# Free energy and metastable states in the square-lattice $J_1$-$J_2$ Ising model

**Veniamin A. Abalmasov**

Institute of Automation and Electrometry SB RAS, 630090 Novosibirsk, Russia

abalmasov@iae.nsc.ru

## Abstract

We calculate the free energy as a function of polarization for the square-lattice $J_1$-$J_2$ Ising model for $J_2 < |J_1|/2$ using the Random local field approximation (RLFA) and Monte Carlo (MC) simulations. Within RLFA, a metastable state with zero polarization is present in the ordered phase. Moreover, the free energy calculated within RLFA indicates a geometric slab-droplet phase transition at low temperature, which cannot be predicted by the mean field approximation. In turn, free energy calculations by definition for finite-size samples using MC simulations reveal metastable states with a wide range of polarization values, the origin of which we discuss. The calculations also reveal additional slab-droplet transitions (at $J_2 > 0.25$). These findings enrich our knowledge of the $J_1$-$J_2$ Ising model and the RLFA as a useful theoretical tool to study phase transitions in spin systems.

# 1 Introduction

The square-lattice $J_1$-$J_2$ Ising model is one of the minimal extension of the standard Ising model, in which the coupling $J_1$ between nearest neighbors is complemented by the coupling $J_2$ between diagonally next-nearest neighbors. The properties of this model are of both fundamental and practical interest, in particular, since its quantum Heisenberg counterpart is relevant to the antiferromagnetism in the parent compounds of the cuprate and pnictide families of high-temperature superconductors [1–3]. Indeed, recent state-of-the-art numerical calculations [4–14] confirm earlier findings [15–22] that diagonal interactions are important in describing the available experimental data for these compounds. Magnetic frustration due to the $J_2$ coupling leads to the quasi-degeneracy of the ground state [6, 19, 20] and possibly to a qunatum spin liquid state at $J_2$ close to $|J_1|/2$ [23, 24].

We recently highlighted the existence of metastable states with arbitrary polarization in the square-lattice $J_1$-$J_2$ Ising model for $J_2 \in (0, |J_1|)$ using Monte Carlo (MC) simulations, which was further supported by simple microscopic energy considerations [25]. For the ferromagnetic ground state, i.e. for $J_1 < 0$ and $J_2 < |J_1|/2$, these states are rectangles with polarization opposite to the surrounding, briefly considered much earlier in [26,27]. Significantly, the Random local field approximation (RLFA) [28], also applied in [25], points to a metastable state with zero polarization in the same $J_2$ coupling range, thus reflecting the appearance of microscopic metastable states, which seems impossible for mean field approximations (MFA). Note that the above states differ from the metastable states of the standard Ising model, consisting of straight stripes, into which a system with zero polarization, when applying the single-spin flip MC algorithm and periodic boundary conditions, relaxes after quenching only in about 30% of cases and only in the absence of an external field [29–31].

Polarization-dependent (called restricted or Landau) free energy $F(m)$, considered in the framework of Landau's phenomenological theory of phase transitions [32], also provides information on metastable states (including those in an external field) and can be used to calculate the relaxation rate of the system to the ground state via the Landau-Khalatnikov equation [33] (see, e.g., [34] for such calculations in ferroelectrics). It should be noted, however, that for short-range interactions, the restricted free energy obtained within the MFA differs qualitatively from the free energy calculated exactly or using the MC method for finite-size samples [35–37]. In the former case, below the phase transition temperature, the free energy takes into account only homogeneous states inside the two-phase (spins up and down) coexistence region and, as a consequence, is a double-well shaped function of polarization. In the latter case, all inhomogeneous states contribute to $F(m)$. Thus, at a temperature close to zero, the barrier between two minima with opposite polarization is determined by the interface energy between two large domains and is proportional to the sample size $L$. Relative to the total energy, proportional to the number of spins $N = L^2$, it vanishes in the thermodynamic limit [38, 39]. It was shown that, despite the loss of detailed information about microscopic spin configurations, $F(m)$ can be harnessed to well reproduce the MC polarization dynamics of the Ising model in good agreement with the droplet theory [40]. These ideas were further developed in [41–44]. The temperature dependence of the free energy barrier in the $J_1$-$J_2$ Ising model, but only in three dimensions, was analytically estimated in [27] in connection with domain growth and shrinking after low-temperature quenching.

Here we calculate the restricted free energy $F(m)$ for the square-lattice $J_1$-$J_2$ Ising model in the framework of RLFA, exactly for a square sample size $L = 6$ and using the MC method for $L = 10$. We pay special attention to the metastable states, which appear in this model and were studied earlier in [25], and explore how they are reflected in the free energy. The features of the geometric slab-droplet phase transition in the free energy calculated by both methods are also briefly discussed.

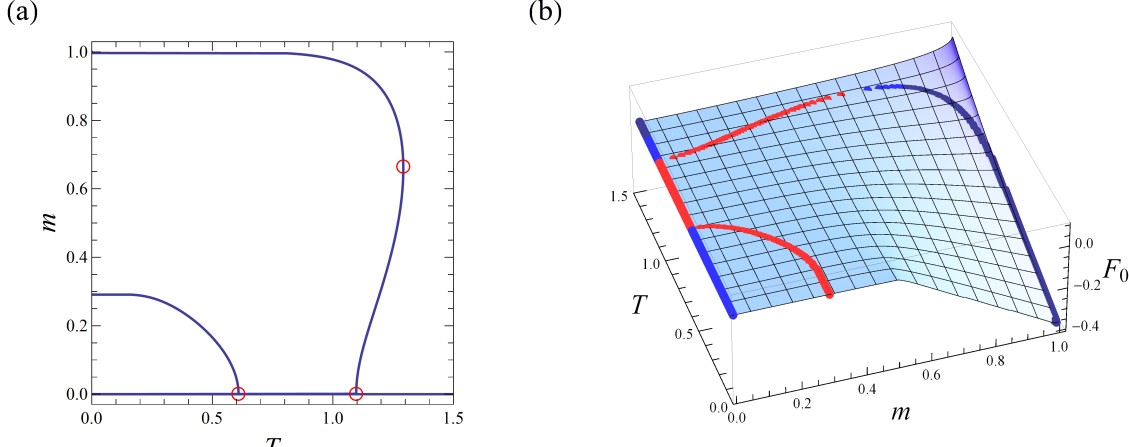

Figure 1: (a) The RLFA solution for $J_2 = 0.3$ (see also Fig. 2 in [25] for different values of $J_2$). Red circles define temperatures $T_0 < T_1 < T_2$. (b) Restricted free energy $F_0(m)$ within RLFA for $J_2 = 0.3$ as a function of temperature. Red points correspond to the local maximum of $F_0(m)$ at each temperature (a barrier), dark blue points correspond to its global minimum (stable states), and light blue points correspond to its local minimum (metastable states), which are zoomed in on in Fig. 2a. Metastable states with zero polarization appear at temperatures from zero to $T_0 \approx 0.6$ and from $T_1 \approx 1.1$ to $T_c < T_2$ ($T_2 \approx 1.26$), at which a first order phase transition occurs.

## 2  Model

The square-lattice $J_1$-$J_2$ Ising model Hamiltonian reads

$$H = J_1 \sum_{\langle i,j \rangle} s_i s_j + J_2 \sum_{\langle\langle i,j \rangle\rangle} s_i s_j - \sum_i h_i s_i, \tag{1}$$

where each spin takes the value plus or minus 1. The sums are over nearest $\langle i, j \rangle$ and diagonal next-nearest $\langle\langle i, j \rangle\rangle$ neighbors, as well as over each spin coupled to the external field $h_i$ at its position. In what follows, we set the values of the coupling constants values $J_1 = -1$ and $J_2 < 1/2$, which correspond to the ferromagnetic ground state (the case $J_2 > 1/2$ with a striped antiferromagnetic ground state is similar in many aspects, but has a more complex spin topology and will be considered separately). Note that the model is invariant with respect to the simultaneous change of the sign of $J_1$ and the replacement of homogeneous polarization with Néel checkerboard one, corresponding to the antiferromagnetic order of the parent compounds of cuprate superconductors [45].

## 3  Random local field approximation

RLFA is based on the exact formula for the average spin [28, 46]:

$$\langle s_i \rangle = \langle \tanh \beta(h_i^s + h_i) \rangle, \tag{2}$$

where $\beta = 1/T$ is the inverse temperature in energy units. The local field, $h_i^s = -\sum_j J_{ij} s_j$, acting on the spin $s_i$ is caused by all spins $s_j$ coupled with it.

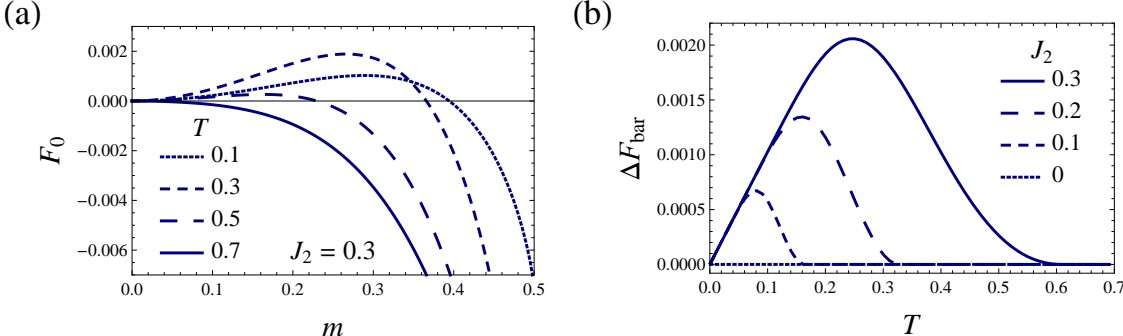

Figure 2: (a) Restricted free energy $F_0(m)$ within RLFA for $J_2 = 0.3$ and polarization limited by $m \in (0, 0.5)$ to show the appearance at low temperature of a barrier at $m \neq 0$ whose height first increases and then decreases as the temperature approaches zero. (b) The barrier height $\Delta F_{\text{bar}} = F_{\text{bar}} - F(0)$ for the metastable state at $m = 0$, which appears in the restricted free energy $F(m)$ calculated within RLFA, as a function of temperature $T$. Only $J_2 < 0.31$ are considered when $T_0 < T_1$ (for temperatures definition see Fig. 1a), since these two temperatures become undefined at larger $J_2$ [25].

The brackets in Eq. (2) correspond to thermal averaging, which is performed with probability distribution [28,47]:

$$P = \prod_j (1 + m_j s_j)/2, \tag{3}$$

where the product is taken over all spins $s_j$ coupled to $s_i$, and $m_j = \langle s_j \rangle = m e^{i\mathbf{q}\mathbf{r}_j}$ is the thermally averaged polarization at position $\mathbf{r}_j$, the variation of which in space is determined by the propagation vector $\mathbf{q}$. Here we consider only homogeneous polarization $m$ and external filed $h$, which corresponds to $\mathbf{q} = (0, 0)$. Note that Eq. (3) implies that, within RLFA, the fluctuations of each spin are considered independent.

Eq. (2) follows from equating to zero the derivative of the restricted free energy $F(m)$ [40], which corresponds to thermodynamic equilibrium at a fixed value of polarization $m$ [48]. To obtain the correct dependence of $F(m)$ on the external field $h$, we rewrite Eq. (2) in the form $\partial F / \partial m = f(m) - h$, integration of which yields $F(m) - F(0) = \int_0^m f(m) dm - hm$. Although $F(0)$ depends on temperature, this is of little interest to us and for convenience we choose $F(0) = 0$ at each temperature and define $F_0(m) = \int_0^m f(m) dm - hm$.

The RLFA solution and the restricted free energy $F_0(m)$ calculated in this way for $J_2 = 0.3$ are shown in Fig. 1. The calculated free energy indeed points to the metastable state with $m = 0$, discussed in [25], which we have zoomed in on in Fig. 2a. With decreasing temperature, a barrier appears at $T_0 \approx 0.6$ near $m = 0$, see Fig. 1a. Then its height first increases and its position shifts up to $m \approx 0.29$, after which, at a temperature slightly less then $J_2$, the height begins to decrease linearly in $T$ to zero, see Fig. 2b. The maximum barrier height of about 0.002 is close to the estimate in [25] based on the value of the coercive field. In Fig. 2b, the barrier height for various values of $J_2$ is shown. At $J_2 \approx 0.31$ we have $T_0 = T_1$, where $T_1$ is the highest temperature at which $F(m)$ is maximum at $m = 0$, see Fig. 1. For larger $J_2$, these two temperatures are not defined [25], and the unstable RLFA solution $m = 0$ extends from 0 to the critical temperature $T_c < T_2$, where $T_2$ is the temperature below which the minimum of $F(m)$ at $m \neq 0$ appears. It should be noted that within RLFA the transition turns out to be first order for $0.25 \lesssim J_2 \lesssim 1.25$ [25], while recent more accurate calculations narrow this interval to a small region around $J_2 = 0.5$ [49–52].

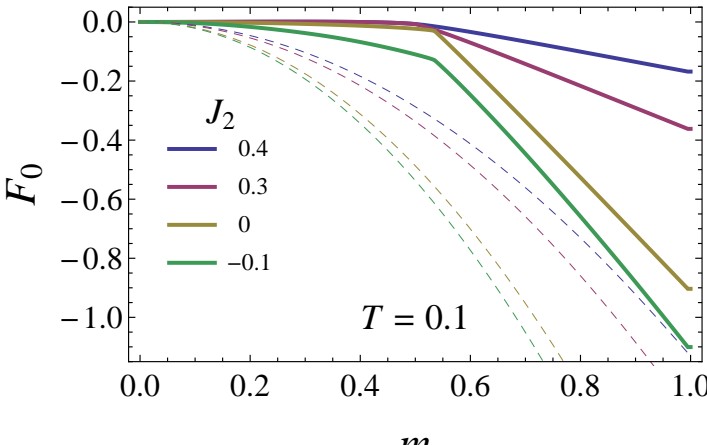

Figure 3: Restricted free energy $F_0(m)$ calculated within RLFA (solid lines) for several values of $J_2$ at temperature $T = 0.1$, which shows a kink at magnetization around $m = 0.5$. The dashed lines are the MFA free energy.

At low temperature, the RLFA restricted free energy shows a kink at a polarization value $m_c \approx 0.5$ for $J_2 = 0.3$, see Fig. 1b. This kink corresponds to a polarization for which the most likely configuration changes from a slab (for $m < m_c$) to a droplet (for $m > m_c$) [40]. For $J_2 = 0$, the RLFA predicted critical polarization $m_c \approx 0.53$ is close to the exact value $m_c = 0.5$ [53], see Fig. 3. In general, this effect is called geometric phase transition and is present in finite-size systems when periodic boundary conditions are used in the simulation [53,54]. For the two-dimensional Ising model, it was thoroughly studied by the Monte Carlo method used to calculate the free energy in [55,56]. We emphasize that RLFA is able to predict the geometrical phase transition, in contrast to MFA. We have also checked that even the four-spin cluster approximation, formulated as in [57], does not predict this transition, despite the good accuracy of the approximation in describing ferroelectric phase transitions [58–61]. It is also worth noting that the RLFA free energy becomes flat, signifying the absence of a phase transition, in the limit $J_2 \rightarrow 0.5$ [25].

# 4 Exact values of restricted free energy

Now we want to compare the above results with the free energy obtained by definition,

$$F(M) = -T \log \sum_E g(M, E) \exp(-E/T), \tag{4}$$

where the sum is over all possible energies $E$ and $g(M, E)$ is the density of states with total spin $M = mN$ and energy $E$ for $N$ spins.

For small samples, the sum in (4) can be computed exactly. For a square sample of size $L = 6$, yielding the total number of spins $N = 36$, the result for $J_2 = 0.3$ is shown in Fig. 4. In all calculations we apply periodic boundary conditions. The critical temperature for $L = 6$ is equal to $T_c = 1.67$, while for $L = 100$ (which practically corresponds to an infinite sample size) it is $T_c = 1.26$ [25]. Configurations that contribute to the free energy $F(M)$ at zero temperature (i.e. have the lowest energies) for different values of $M$ are shown in Fig. 5. At zero temperature, at $m \lesssim 0.5$, the free energy is flat for $0 < J_2 < 0.5$, Fig. 6a. The pits, also mentioned in [40], are due to spin configurations with completely flat interface between two slabs with opposite spin

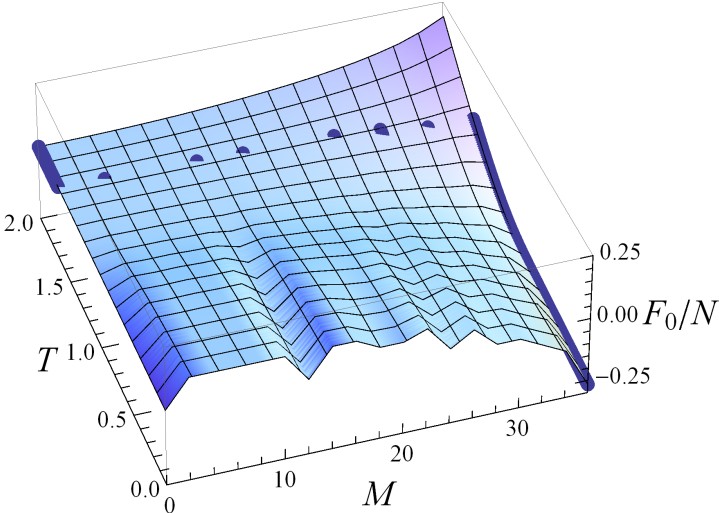

Figure 4: Restricted free energy per spin, $F_0/N$, calculated exactly by Eq. (4) at $J_2 = 0.3$ for sample size $L = 6$ as a function of total spin $M$ and temperature $T$. The free energy is defined only for integer even values of $M$ and linearly interpolated between them. For ease of comparison with the RLFA results in Fig. 1, we set here $F = 0$ at $M = 0$ at each temperature. Dark blue points correspond to the global minimum of $F_0/N$ at each temperature.

directions, see configurations with $M = 0$ ($M_0$) and $M = 12$ ($M_{12}$) in Fig. 5. When a spin flips on this interface (configurations $M_2$ and $M_{14}$ in Fig. 5) the energy increases by $4J_1$. When the last spin in the row flips, the energy decreases by this value (see transition $M_{10} \rightarrow M_{12}$ in Fig. 5 and Fig. 6a). The distance between two neighbor pits of $F(M)$ at $m \lesssim 0.5$ is equal to $2L$, since any spin flip changes the total moment by 2.

At $m \gtrsim 0.5$, the configurations that contribute to the free energy $F(M)$ at zero temperature correspond mainly to the droplet (starting from configuration $M_{18}$ in Fig. 5), but depend on $J_2$, as indicated in the bottom row of Fig. 5. The latter also affects the dependence of the energy barrier height, $\Delta F_M = F_M - F_{M-2}$, on $J_2$ (Fig. 6a). As shown in Fig. 7a, for $M = 22$ at $J_2 < 0.33$ and for $M = 26$ at $J_2 < 0.25$, the barrier height is $4J_2$ and is determined by the spin flip at the corner of the droplet. Note that the exact values of $J_2$ are equal to 1/3 and 1/4 and follow from the energy ratio of the different configurations $M_{22}$, $M_{24}$, and $M_{26}$ in Fig. 5. At larger values of $J_2$ for both values of $M$, there is a reverse transition to the slab phase and then back again, as can be seen in the bottom row of Fig. 5. When a spin flips on a side of the droplet whose length $D > 1$, the energy does not change until the last spin on the side flips, then the energy decreases by $\Delta E = 4(J_1 + J_2)$ (see transitions $M_{22} \rightarrow M_{24}$ at $J_2 < 0.25$ and $M_{26} \rightarrow M_{28}$ at $J_2 < 0.33$ in Fig. 5 and Fig. 6a). For $D = 1$, $\Delta E = 4(J_1 + 2J_2)$, which is valid for transitions $M_{30} \rightarrow M_{32}$ and $M_{32} \rightarrow M_{34}$ in Fig. 6a. At $J_2 \leq 0$, we see only decrease in free energy with $M$ for the droplet phase in Fig. 6a.

At higher temperatures, other higher energy configurations in addition to those shown in Fig. 5 contribute to the partition function in Eq. (4) for each value of $M$. This affects the dependence of the above discussed energy barriers on temperature, which for $J_2 = 0.3$ is shown in Fig. 8a. The metastable state barrier at $M = 22$ disappears at $T \approx 0.65$, which is close to the corresponding temperature $T_0 \approx 0.6$ from the RLFA solution (see Fig. 1). Note that at this value of $J_2$ the barrier at $M = 26$ is determined by the slab-droplet transition and not by the metastable state (see Fig. 5 and Fig. 7a).

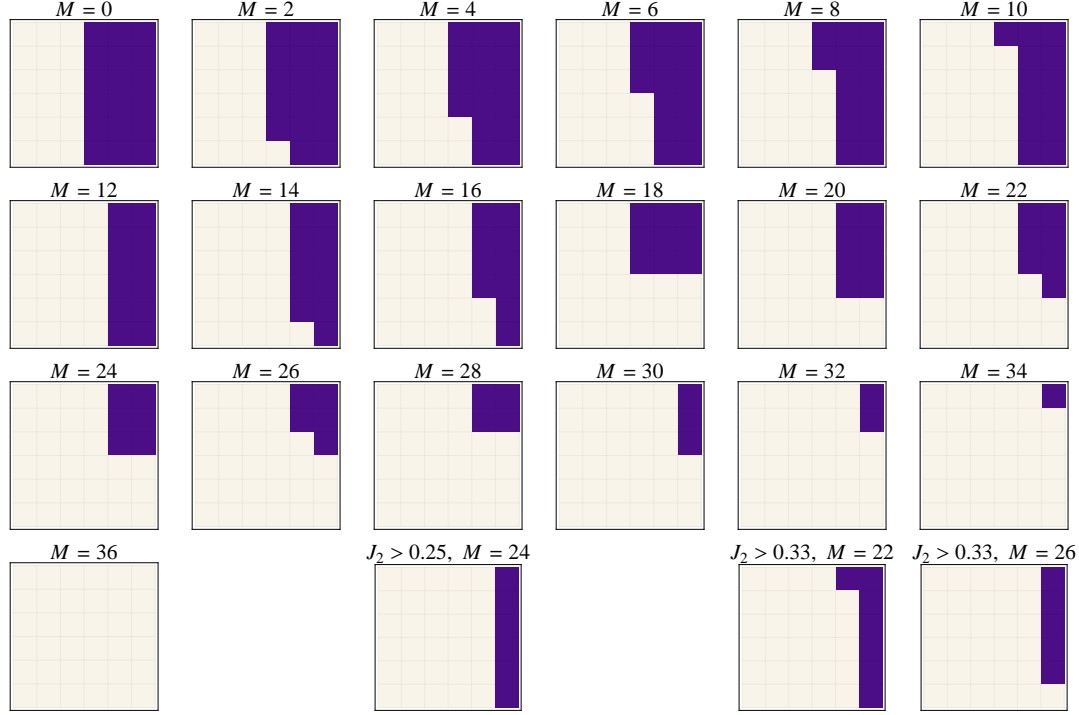

Figure 5: Configurations that contribute to the restricted free energy $F(M)$ at zero temperature, i.e. have minimal energies, for all possible values of the total spin $M$ at $L = 6$. The configurations that change for $J_2 > 0.25$ and $J_2 > 0.33$ are shown separately in the bottom row (they affect the dependence of the energy barrier $\Delta F_M$ on $J_2$ shown in Fig. 7).

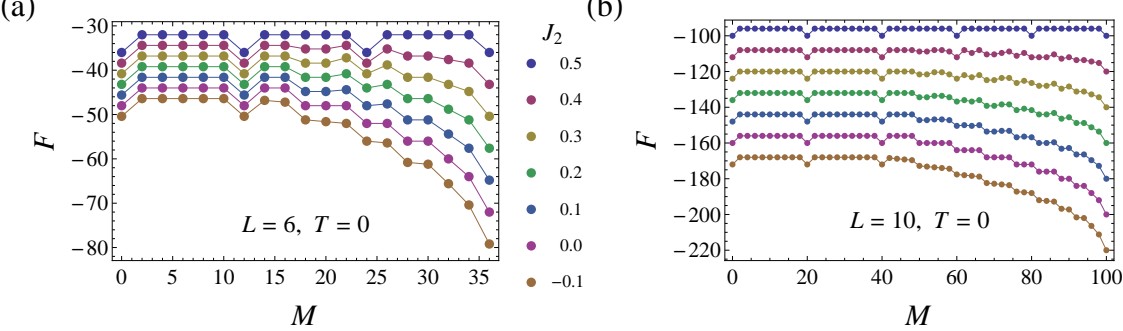

Figure 6: Restricted free energy $F$ as a function of total spin $M = mL^2$ at zero temperature $T = 0$, calculated according to Eq. (4) for several values of $J_2$ (listed in the legend, valid for both plots), (a) exactly for the sample size $L = 6$, (b) using Monte Carlo method for $L = 10$. The solid lines provide guides to the eye.

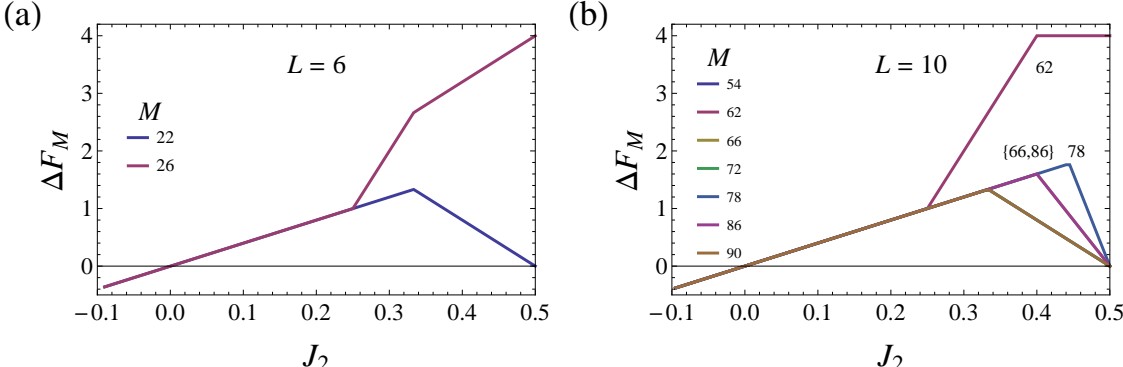

Figure 7: Restricted free energy barrier height, $\Delta F_M = F_M - F_{M-2}$, as a function of $J_2$ for several values of the total spin $M$ at zero temperature. (a) The sample size is $L = 6$. (b) The sample size is $L = 10$. Lines for different values of $M$ overlap. The lines corresponding to $M = 62, 78$ and overlaping $\{66, 86\}$ are marked separately.

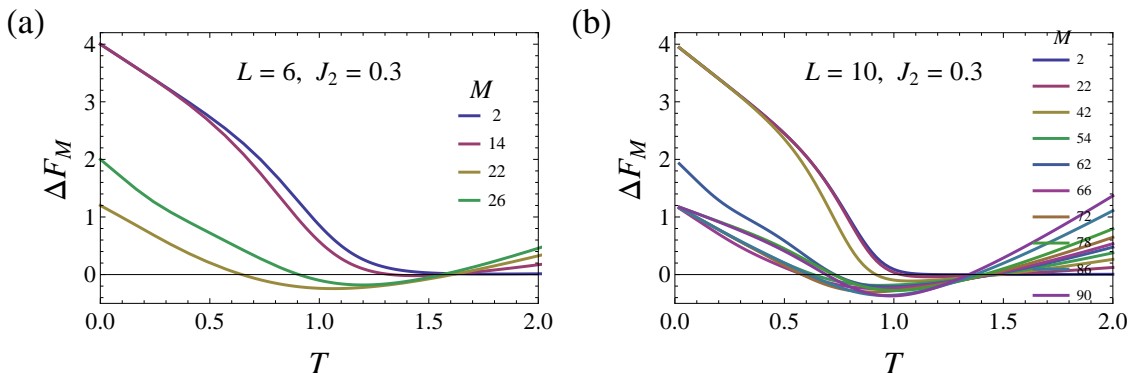

Figure 8: Restricted free energy barrier height, $\Delta F_M = F_M - F_{M-2}$, as a function of temperature for several values of the total spin $M$ and $J_2 = 0.3$. (a) The sample size is $L = 6$. (b) $L = 10$. The three upper curves correspond to $M = 2$, 22 and 42, and in the middle is $M = 62$.

## 5 Monte Carlo simulations

For larger square samples, with $L = 7$ and $L = 8$, the free energy from Eq. (4) can only be calculated using supercomputers, given the large number of $2^N$ configurations for $N$ spins. Alternatively, it can be calculated approximately with sufficiently high accuracy using the Monte Carlo method. We use the Wang-Landau algorithm [62–64], which has proven to be very efficient for this purpose at low temperature. It consists in performing a random walk in polarization and energy space to extract an estimate for the density of states $g(M, E)$ that gives a flat histogram.

Using the Wang-Landau algorithm, we reproduce the exact results for $L = 6$ with high accuracy and obtain similar result for $L = 10$, see Fig. 6b and Fig. 9, where the free energy barriers at $m > m_c$ are clearly visible at low temperature and $J_2 = 0.3$. For $J_2 = 0$, the calculated free energy at zero temperature, Fig. 6b, agrees with [40]. I note that the free energy for larger samples could also be calculated, as was done, for example, in [40] for $J_2 = 0$. However, for the purposes of this article, namely to show how metastable states are reflected in the free energy, the size $L = 10$ seems optimal, and all metastable states are clearly

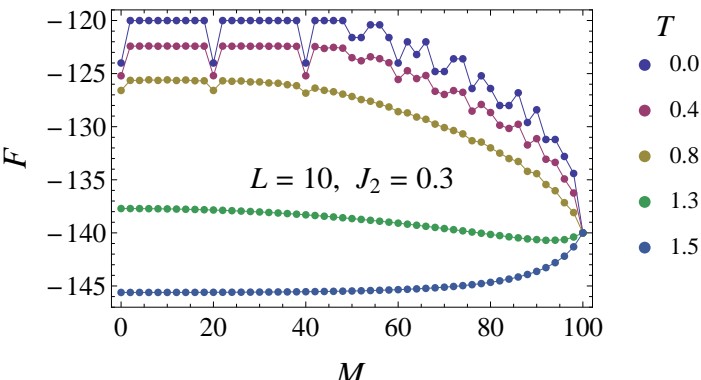

Figure 9: Restricted free energy $F$ as a function of total spin $M$ for $J_2 = 0.3$ calculated by the MC method at several temperatures below and above the phase transition. The sample size is $L = 10$.

visible and convenient for analysis.

The barrier heights for metastable states at $m > m_c$ correspond to the spin flip at the corner of the droplet and are equal to $4J_2$ for $J_2 < 0.25$ (Fig. 7b). For $J_2 > 0.25$, this dependence changes due to additional slab-droplet and vice versa transitions, as in the case of $L = 6$. The dependence of these barriers on temperature is shown in Fig. 8b. It is almost linear and the barriers disappear in the temperature range from approximately 0.6 to 0.7, which is close to the RLFA predicted $T_0$ in Fig. 1. We verified that the linear dependence of barriers height on temperature and their disappearance at a temperature close to $T_0$, obtained within RLFA, are also valid for other values of $J_2$. Note that the linear dependence on $T$ at low temperature follows directly from the definition of the free energy $F = U - TS$, where $U$ is the energy and $S$ is the entropy.

## 6  Discussion

The primary goal of this work was to confirm, by calculating the restricted free energy, the presence of metastable states in the $J_1$-$J_2$ Ising model, recently found in the RLFA solution and MC simulations of low-temperature quenching [25]. The restricted free energy $F(m)$ as a function of polarization calculated within RLFA indeed shows local minima at zero polarization at low temperature for $J_2 > 0$, see Fig. 1 and Fig. 2, thus indicating a metastable state.

At the same time, the exact calculations of $F(m)$ for a small sample size $L = 6$ (Fig. 4 and Fig. 6a) and MC simulations for $L = 10$ (Fig. 6b and Fig. 9) indicate local minima corresponding to metastable states with various values of polarization. Some of them, at $m \lesssim 0.5$, are due to long stripes with an activation energy of $4J_1$ of a spin flip on a flat domain boundary (Fig. 5). In the standard Ising model, the system can become stuck in these states with a final polarization following a Gaussian distribution after zero-temperature quenching from an initially random configuration with zero polarization [30].

Metastable states at $m \gtrsim 0.5$ are caused by droplet-shaped domains with an activation energy of $4J_2$ of a spin flip in their corner, at least at $J_2 < 0.25$ for both sample sizes $L = 6$ and $L = 10$ (Fig. 7). It is interesting whether this value of $J_2$ holds true for larger sample sizes. At $J_2 > 0.25$, the dependence of the barrier height on $J_2$ for some metastable states changes. Our exact energy calculations for a sample size of $L = 6$ show that for $J_2 > 1/4$ and then for $J_2 > 1/3$ the sequence of minimal energy configurations for increasing total spin $M$ changes (see Fig. 5) and a return occurs into the slab phase at certain values of $M$, which

may be important for some applications. Indeed, the importance of the geometrical slab-droplet transition for various physical situations, including the dewetting transition between hydrophobic surfaces, was highlighted in [54].

Although the zero polarization of the metastable state and the low height of the barrier proportional to the temperature near zero (see Fig. 2) is not exactly what follows from MC calculations, where the barrier heights are much higher and decrease with temperature (see Fig. 8), the fact of even a rough indication of the metastable state by RLFA is very valuable.

Another valuable RLFA prediction that turns out to be quite accurate is the geometric slab-droplet phase transition at low temperature (Fig. 1b and Fig. 3). The reason why RLFA is so effective in this situation, in our opinion, is that by definition it takes into account the local field due to all possible configurations of spins interacting with the central spin, not just the mean field. The probability of these configurations, in turn, is determined by the mean spin.

The distance between striped metastable states along the $M$ axis (at $m \lesssim 0.5$) is equal to $2L$ and, as a result, their number is proportional to $L$. For droplet metastable states (at $m \gtrsim 0.5$), the distance is determined by the droplet size, which becomes smaller as $M$ increases. Thus, one can expect that the number of droplet metastable states scales as $L^2$ and they are distributed along the $M$ axis much more densely. This is confirmed by our MC simulations in Figs. 6, 7, and 9. This could be the reason why, during low-temperature quenching from high temperature in an external field, the system is not captured into striped metastable states in the standard Ising model [30], but gets stuck in droplet metastable states with finite polarization when $J_2 \in (0, 1/2)$ [25]. Note that the polarization of such a final state turns out to be about 0.5 at zero temperature [25], which is close to the slab-droplet phase transition, from where droplet metastable states begin to appear as $M$ increases (Fig. 9). However, the polarization after quenching (from high temperature) sharply decreases with final temperature [25] and does not correspond to the critical polarization $m_c$ of the slab-droplet transition in the standard Ising model, the temperature dependence of which resembles the equilibrium polarization [53].

It should be noted here that the metastable states into which the system relaxes after low-temperature quenching in [25] are not exactly the same as shown in Fig. 5 and which determines the free energy at zero temperature. The energy of the former is much higher, and the system is more likely to get stuck in them, relaxing in energy during quenching on the way to thermal equilibrium. Metastable states like in Fig. 5 can in principle be reached after quenching at non-zero temperature after a sufficiently long relaxation time and domain coarsening, with a higher probability for those closer to the equilibrium polarization. At the same time, any of these states will be reached inevitably if the total spin is conserved during quenching, as in the Kawasaki [65] two-spin exchange algorithm, which is relevant for models describing transport phenomena caused by spatial inhomogeny such as diffusion, heat conduction, etc.

Finally, we will mention some recent advances in the experimental observation of metastable states using sub-picosecond optical pulses, which we believe can be applied to reveal metastable states discussed here and in [25]. For instance, in the quasi-two-dimensional antiferromagnet $Sr_2IrO4$, a long-range magnetic correlation along one direction was converted into a glassy condition by a single 100-fs-laser pulse [66]. Atomic-scale $PbTiO_3/SrTiO_3$ superlattices, counterpoising strain and polarization states in alternate layers, was converted by sub-picosecond optical pulses to a supercrystal phase in [67]. In a layered dichalcogenide crystal of $1T$-$TaS_2$, a hidden low-resistance electronic state with polaron reordering was reached as a result of a quench caused by a single 35-femtosecond laser pulse [68]. See also the references to relevant superconducting and magnetic materials with next-nearest-neighbor interactions mentioned in Introduction and [25].

## 7 Conclusion

In conclusion, we calculated the restricted free energy $F(m)$ as a function of polarization $m$ for the square-lattice $J_1$-$J_2$ Ising model (at $J_2 < |J_1|/2$) within RLFA and using the MC method. Both approaches indicate the appearance of metastable states at low temperature, corresponding to local minima of $F(m)$ along the $m$ coordinate. The zero-polarization metastable state predicted by RLFA reflects the true metastable states with various polarization values at $m \gtrsim 0.5$ that appear in our exact calculation and MC simulations of the restricted free energy. We show that RLFA predicts the slab-droplet phase transition for the $J_1$-$J_2$ Ising model as a kink in the polarization dependence of $F(m)$ at low temperature. Exact calculations of $F(m)$ for a sample size of $L = 6$ reveal also additional slab-droplet transitions at $J_2 > 0.25$. We believe, easy-to-use RLFA can help reveal the presence of metastable states and geometrical phase transitions in more complex systems, e.g., with site or bond disorder and spin tunneling in a transverse field.

## Acknowledgments

I thank B.E. Vugmeister for many useful discussions. The Siberian Branch of the Russian Academy of Sciences (SB RAS) Siberian Supercomputer Center is gratefully acknowledged for providing supercomputer facilities.

**Funding information**    I acknowledge the support by the Ministry of Science and Higher Education of the Russian Federation (Grant No. 1023032300239-8).

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
