# Peer review of "Free energy and metastable states in the square-lattice J1-J2 Ising model"

_SciPost Physics_

## Round 1 · Referee Report · Pranay Patil · 2024-1-24

Strengths
1- Use of a sophisticated mean field theory based ansatz to understand a known problem of meta-stability in a toy model.
2- Comparison with Monte Carlo simulations provided.
Weaknesses
1- Results do not appear to add obvious value over a recent paper by the same author (https://journals.aps.org/pre/abstract/10.1103/PhysRevE.107.034124), where the same problem has been tackled using the same method.
2- The indicators for meta-stability are indirect and the results are confirmed using differences in free energy based on expectations of slab versus droplet pictures. This is not a smoking gun signature. I would suggest looking at correlation functions (if possible) in both RLFA and MC simulations to confirm the presence of these spatial structures.
3- The discussion about the ability of RLFA to capture the meta-stable states is not sufficient, and requires added details which can convince the reader as to the added expression ability provided by RLFA.
4- Monte Carlo simulations are not performed at a sufficient size. If they are scaled up, direct snapshots can be shown to indicate the expected spatial structures.
Report
The author has considered the problem of meta-stability created by competing interactions in an Ising model on a square lattice. The method used here is a mean field ansatz which is capable of treating nearest neighbor correlations. Stable spatial patterns are argued for using differences in free energy at fixed magnetization. A comparison to Monte Carlo results is provided. As differences in free energy are employed, the inference of particular spatial patterns is indirect and it would be beneficial is something more direct (such as correlation functions) could be provided. The central claim is the ability of the particular flavor of mean field ansatz to represent various meta-stable spatial patterns, and this is not adequately supported by the data presented
Requested changes
1- A more detailed explanation of the RLFA procedure and its ability to encode spatial patterns is required.
2- Direct evidence (such as correlation functions) should be provided using both RLFA and MC simulations. This is definitely possible for the latter. For RLFA, this may be a challenging task, but it would definitely strengthen the claim the author wishes to make.
Author: Veniamin Abalmasov on 2024-05-02 [id 4461]
(in reply to Report 1 by Pranay Patil on 2024-01-24)
Dear Pranay Patil,
Thank you for noting the strength of the paper. Let me comment on your criticism.
Weaknesses 1- Results do not appear to add obvious value over a recent paper by the same author (https://journals.aps.org/pre/abstract/10.1103/PhysRevE.107.034124), where the same problem has been tackled using the same method.
In many cases, free energy is the starting point for studying phase transitions, stable and metastable states, whereas in our previous paper metastable states were studied by the equation of state within RLFA and low-temperature quenching simulations using MC. In this work, we look at these metastable states from the free energy perspective. In the case of RLFA, the calculated Landau free energy shows explicitly a local energy minimum corresponding to a metastable state with zero polarization. In addition, a kink in the Landau free energy within RLFA was discovered, associated with a geometrical phase transition. At the same time, the calculated restricted free energy only reveals the lowest energy metastable states as we discuss in the paper, which is nevertheless also a good result.
2- The indicators for meta-stability are indirect and the results are confirmed using differences in free energy based on expectations of slab versus droplet pictures. This is not a smoking gun signature. I would suggest looking at correlation functions (if possible) in both RLFA and MC simulations to confirm the presence of these spatial structures.
RLFA is an approximation, and its prediction of a metastable state with zero polarization is only an indication of the possible existence of metastable states in the model. This prediction is valuable, even if it is very approximate. Metastable states revealed using the restricted free energy calculated for a finite-size sample have the lowest possible energy at a given value of polarization. Thus, they cannot directly reproduce the numerous high-energy metastable states that arise after low-temperature quenching. However, their proper combinations can. I now mention this in Sec. 6, paragraph 5. I have also added a discussion of the restricted free energy, which depends on nearest neighbor correlations (Sec. 4, the last two paragraphs and Sec. 5, the last paragraph, together with Figs. 5, 6 and 16).
3- The discussion about the ability of RLFA to capture the meta-stable states is not sufficient, and requires added details which can convince the reader as to the added expression ability provided by RLFA.
I have now added a discussion of how geometric phase transition appears within RLFA (Appendix B.2). This is due to the statistical contribution of different local fields. The same is true for predicting a metastable state (as noted at the end of App. B.2).
4- Monte Carlo simulations are not performed at a sufficient size. If they are scaled up, direct snapshots can be shown to indicate the expected spatial structures.
A larger sample size will result in a larger maximum value of the total spin M and more metastable states in the plot of F vs M. But these metastable states will have the lowest energy for a given M, having just two domains (the same as for smaller samples). I now note this in Sec. 5, second paragraph. I have now also added MC simulations for sample size L = 14 for comparison in Appendix B.2, Fig. 15.
The Monte Carlo simulation that I use here gives an estimation of the density of states for given value of the total spin M and energy E (see Sec. 5, first paragraph). It does not provide snapshots of metastable states, as in the case of low-temperature quenching.
Requested changes 1- A more detailed explanation of the RLFA procedure and its ability to encode spatial patterns is required.
It has now been added to the end of Appendix A.2 as noted above.
2- Direct evidence (such as correlation functions) should be provided using both RLFA and MC simulations. This is definitely possible for the latter. For RLFA, this may be a challenging task, but it would definitely strengthen the claim the author wishes to make.
The restricted free energy as a function of nearest-neighbor correlations is now discussed at the end of Sec. 4, the last two paragraphs (see also Figs. 5, 6) and Sec. 5, the last paragraph (also Fig. 16 in Appendix B.3). For RLFA, such calculations seem very complicated if possible (they would probably point to the lowest energy metastable state, as is the case with the restricted free energy in MC calculations).
Author: Veniamin Abalmasov on 2024-05-02 [id 4463]
(in reply to Report 3 by Hiroshi Watanabe on 2024-02-09)Dear Hiroshi Watanabe,
Thank you for your interest in the paper. Below I respond to your comments.
I have now reorganized the manuscript and moved some of the details and descriptions into Appendices. I hope this helps reading and understanding.
Now that some details have been transferred to the Appendices, the two 3D figures (Fig. 1 and former Fig. 4, which is now Fig.3) are closer to each other for comparison. It is difficult to place these two figures side by side, since they were obtained using different techniques, successively described in the article. I also note that the difference between the Landau free energy within mean field approximations and the restricted free energy obtained for finite-size samples is emphasized in Sec. 1, paragraph 3. The figures may look similar, but in essence the functions are different.
Now these temperatures are shown explicitly in Fig.1a.
Now these figures are in Appendices (Fig. 8 and Fig. 14). It is difficult to place them one after another from the point of view of the logic of presentation.
In principle, one can trace each barrier in this plot (former Fig.8(b)), but the problem is that there are a lot of them. Fig. 8(b) compared to Fig. 8(a) (now Fig. 14(b) and Fig. 14(a)) only shows that the barriers behavior does not change much as the sample size increases.
I call the order parameter polarization because it can be applied both to magnetic (magnetization) and electric polarization. I started studying this problem with ferroelectrics, where one deals with (electric) polarization. Now I note this in Sec. 3 after Eq. (3).
The total spin M for finite-size samples takes only integer values that are easy to relate with the corresponding spin configurations. In addition, any single-spin flip changes the total spin by 2. At the same time, the correspondence between polarization m and total spin M is obvious, m = M/N, where N is the number of spins in a sample. That is why I prefer to use total spin M dealing with the restricted free energy for finite-size samples. Now I note this in Sec. 4 after Eq. (4).
The geometric phase transition occurs for free boundary conditions as well. I have now corrected this passage in the manuscript and included the corresponding Fig. 15(b) in Appendix B.2.
I now discuss the derivation of the Landau free energy in more detail in Appendix A.2 for both RLFA and MFA. The ability of RLFA to predict the geometric phase transition is based on averaging over all possible values of the local field (and corresponding local spin configurations, the contribution of which depends on polarization).
These issued have been commented above.

---

## Round 1 · Referee Report · Li-Ping Yang · 2024-2-8

Strengths
very detailed analysis
the self-consistent representation
Weaknesses
The big difference in the range of J2 for the first order phase transition between RLFA and previous numerical calculation, especially the recent results, needs more convincing discussions. Perhaps, the RLFA itself , the analysis of the entropy , or something else, is accountable.
Report
Compared to the relevant paper [20] in the references, the author made a further investigation by RLFA and restricted free energy. As demonstrated, the free energy barrier explained the formation of the metastable states as the temperature and J2 varied.
By a detailed calculation of size 6*6 in fig.5, the author illuminated the slab-droplet geometric phase transition. By RLFA, the non-analytic behaviors signals the phase transition. It is a perspective angle to enrich the understanding for J1-J2 model.
I feel the main context is too long, it would be better to move some figures and relevant discussions to the supplemental materials for the compactness in the article, if possible.
The author have made a self-consistent analysis for the results from RLFA, MC, and restricted free energy. I still wonder the big difference of the range of J2 determining the first order phase transition between previous numerical calculation and the results in this paper. More convincing discussions, e.g. the characters of RLFA , the effect from the entropy, or something else, is to be discussed for the completeness.
In addition, a typo in line #94, less then-> less than.
Requested changes
More convincing discussions about the big difference of J2 regime between RLFA and previous numerical results for the first order phase transition the are needed.
Author: Veniamin Abalmasov on 2024-05-02 [id 4462]
(in reply to Report 2 by Li-Ping Yang on 2024-02-08)
Dear Li-Ping Yang,
Thank you for your interest in the paper and finding its strengths. Below I will respond to your criticism.
Weaknesses The big difference in the range of J2 for the first order phase transition between RLFA and previous numerical calculation, especially the recent results, needs more convincing discussions. Perhaps, the RLFA itself , the analysis of the entropy , or something else, is accountable. Requested changes More convincing discussions about the big difference of J2 regime between RLFA and previous numerical results for the first order phase transition the are needed.
The first-order phase transition range of J2 within RLFA is slightly larger but comparable to the results of cluster mean field approximations (now I mention this in Sec. 3, paragraph 4). These approximations, so to speak, belong to more or less the same class of complexity, just one step beyond the mean field approximation, with their own peculiarities. I am not sure that the accuracy of RLFA can be easily improved in this regard. Other techniques that provide more accurate results require much more computational effort.
I feel the main context is too long, it would be better to move some figures and relevant discussions to the supplemental materials for the compactness in the article, if possible.
I agree with you on this issue. I have now reformatted the manuscript and moved some of the details and descriptions into the Appendices.

---

## Round 1 · Referee Report · Hiroshi Watanabe · 2024-2-9

Strengths
Calculations and analysis were carefully performed.
Weaknesses
Poorly organized and difficult to understand.
Report
The author investigated the free energy of the square-lattice J1-J2 Ising model using the random local field approximation (RLFA). The author compared the prediction by the RLFA with the exact solution and Monte Carlo results for small systems. This paper continued with another article by the same author [25]. This paper aimed to study the metastable states identified in the previous paper in terms of free energy. The free energy calculated by the RLFA indicated the existence of the metastable states, and they were confirmed in the exact calculations and MC simulations. The geometric slab-droplet transition was also investigated.
The subject matter addressed in this paper is somewhat specialized. Still, the calculations and analysis have been carefully performed, and we are willing to publish it with minor modifications.
While the subject matter of this paper is somewhat very specialized, the calculations and analyses were performed carefully. Therefore, I agree to publish the manuscript with minor modifications.
Here are my comments.
(1) Organization of the manuscript
The paper is poorly organized and requires much effort from the reader. It also took a lot of work to understand the correspondence between the figures.
For example, Fig. 1 and Fig. 4 seem to correspond, but it isn't easy to see how they should be compared from the figures.
The symbols T_0, T_1, and T_2 appear without explanation in the caption of Fig. 1, and their definitions appear discretely in the text.
Also, Fig. 2 (a), Fig. 6 and Fig. 2 (b), and Fig. 8 seem to correspond to each other, but it isn't easy to read because it was necessary to refer to several places simultaneously to understand the context.
It is also difficult to extract useful information from Fig. 8(b).
The author should structure the manuscript more clearly for the reader.
(2) The definitions of the order parameters
The polarization m was adopted as the order parameter in the analysis with RLFA, while the total spin M was used as the order parameter for the exact solutions and MC simulations, which makes it difficult to compare the figures.
Also, since this paper deals only with homogeneous polarization, m seems identical to magnetization. Is it correct? For example, the caption of Fig. 3 refers to m as magnetization. Then, why not call all of them magnetization?
It would be easier to see if we use m as the order variable instead of the total spin M in Figs. 5-9. Is there any particular reason to use M?
(3) Comparison with MFA
The author claimed that the RLFA predicted the slab-droplet transition that MFA does not. Also, the author claimed that such geometric phase transition is present in the finite system with periodic boundary conditions.
If I understand correctly, RLFA is a kind of MFA, and the boundary conditions cannot be considered explicitly.
The author should discuss why RLFA could predict the slab-droplet transition, which MFA could not.
Requested changes
1- Modify figures to make it easier for readers to compare.
2- Add the detailed comparison with MFA

---

## Editorial Decision

unknown